# Structure-Function Correlation of Retinal Fibrosis in Eyes with Neovascular Age-Related Macular Degeneration

**DOI:** 10.3390/jcm13041074

**Published:** 2024-02-14

**Authors:** Markus Schranz, Stefan Sacu, Gregor S. Reiter, Magdalena Baratsits, Silvia Desissaire, Michael Pircher, Georgios Mylonas, Christoph Hitzenberger, Ursula Schmidt-Erfurth, Philipp Ken Roberts

**Affiliations:** 1Vienna Clinical Trial Center (VTC), Department of Ophthalmology and Optometry, Medical University of Vienna, 1090 Vienna, Austria; markus.schranz@meduniwien.ac.at (M.S.);; 2Department of Ophthalmology and Optometry, Medical University of Vienna, 1090 Vienna, Austria; 3Center for Medical Physics and Biomedical Engineering, Medical University of Vienna, 1090 Vienna, Austria

**Keywords:** nAMD, fibrosis, multimodal imaging, microperimetry

## Abstract

**Purpose:** To assess retinal function in areas of presumed fibrosis due to neovascular age-related macular degeneration (nAMD), using multimodal imaging and structure-function correlation. **Design:** Cross-sectional observational study. **Methods:** 30 eyes of 30 consecutive patients with nAMD with a minimum history of one year of anti-vascular endothelial growth factor therapy were included. Each patient underwent microperimetry (MP), color fundus photography (CFP), standard spectral-domain-based OCT (SD-OCT), and polarization sensitive-OCT (PS-OCT) imaging. PS-OCT technology can depict retinal fibrosis based on its birefringence. CFP, SD-OCT, and PS-OCT were evaluated independently for the presence of fibrosis at the corresponding MP stimuli locations. MP results and morphologic findings in CFP, SD-OCT, and PS-OCT were co-registered and analyzed using mixed linear models. **Results:** In total, 1350 MP locations were evaluated to assess the functional impact of fibrosis according to a standardized protocol. The estimated means of retinal areas with signs of fibrosis were 12.60 db (95% confidence interval: 10.44–14.76) in CFP, 11.60 db (95% COI: 8.84–14.36) in OCT, and 11.02 db (95% COI 8.10–13.94) in PS-OCT. Areas evaluated as subretinal fibrosis in three (7.2 db) or two (10.1 db) modalities were significantly correlated with a lower retinal sensitivity than a subretinal fibrosis observed in only one (15.3 db) or none (23.3 db) modality (*p* < 0.001). **Conclusions:** CFP, SD-OCT and PS-OCT are all suited to detect areas of reduced retinal sensitivity related to fibrosis, however, a multimodal imaging approach provides higher accuracy in the identification of areas with low sensitivity in MP (i.e., impaired retinal function), and thereby improves the detection rate of subretinal fibrosis in nAMD.

## 1. Introduction

Age-related macular degeneration (AMD) is the leading cause of blindness in industrialized countries with an increasing prevalence due to an aging population and nearly 288 million patients expected worldwide by 2040 [1,2,3].

Between 10–15% of patients develop neovascular or exudative AMD, resulting in retinal edema, hemorrhage, and/or fibrosis with irreversible damage to the neurosensory retina.

Even under optimal treatment with anti-vascular endothelial growth factor (anti-VEGF), patients are exposed to an almost 50% risk of developing fibrosis within the first 2 years, especially in cases with Type 2 macular neovascularization (MNV) [4]. Imbalance of anti-VEGF and connective tissue growth factor (CTGF) [5], RPE disorganization [6], and several immune responses [7] are part of the fibrotic process in neovascular AMD.

The diagnosis of fibrotic tissue or retinal scarring In a clinical or reading center setting is based on color fundus photography (CFP), optical coherence tomography (OCT), and rarely on fluorescence angiography (FA). CFP is currently the gold standard for detecting subretinal fibrosis, however, there are limitations in the ability to differentiate fibrosis from other white-yellowish AMD-related structures such as drusen, exudate, fibrin deposition, or other material mimicking fibrosis [8]. Similar limitations apply to standard OCT B-scans, where fibrotic lesions appear as subretinal or sub-RPE hyperreflective layers but cannot be reliably differentiated from other subretinal hyperreflective material (SHRM) such as fibrin, the MNV complex, and organized blood. The diagnosis of subretinal fibrosis using a conventional OCT device may therefore often be imprecise or even impossible [9]. However, increasing the precision of diagnosing retinal fibrosis is crucial as scarring is still a relevant limiting factor for the visual acuity of patients with nAMD [10,11,12,13]. Furthermore, evaluation of the efficacy of anti-fibrotic agents is challenging since reliable study endpoints are lacking using standard imaging modalities. Reliable in vivo detection and quantification of small amounts of fibrosis are necessary requirements for an assessment of successful therapeutic response. Trials investigating anti-fibrotic substances necessarily require an imaging modality, which allows quantitative objective in vivo detection of retinal fibrosis.

In contrast to standard spectral domain or swept source-OCT images, in which retinal layers are distinguished based on the intensity of backscattered light, polarization sensitive-OCT (PS-OCT) is capable of selectively visualizing different retinal layers based on tissue-specific contrast. It enables the measurement of multiple variables simultaneously: the intensity of the backscattered light (as in conventional OCT imaging), birefringence (measured as retardation and optic axis orientation), as well as the degree of polarization uniformity (DOPU), a quantity related to depolarization [14].

PS-OCT can be used to identify different SHRM components in the retina such as MNV membranes, fibrosis, hemorrhage, or RPE rips by tissue-specific contrast [15,16].

For example, melanin-containing structures within the healthy RPE cell body cause depolarization whereas organized fibers such as collagenous fibers induce birefringence within retinal fibrosis, which can be used to automatically detect fibrosis in PS-OCT B-scans [17,18,19,20,21].

MP evaluates retinal function on a topographic level. Due to its fundus-related-eye-tracking system it can be easily combined with other imaging modalities allowing detailed structure–function assessment [22,23]. Retinal sensitivity is expected to be lower in areas of retinal fibrosis compared to healthy retina. MP may therefore be used to detect impaired retinal sensitivity as an independent functional correlate of subretinal fibrosis, however, data on the structure–function correlation in eyes with retinal fibrosis due to nAMD is limited. Recently, a review paper on fibrosis in nAMD by Cheong et al. concluded that more studies involving multimodal imaging are necessary to clarify our understanding of retinal fibrosis, as each imaging modality has its advantages and disadvantages for imaging fibrosis [11]. Furthermore, while there is plenty of data evaluating visual acuity in eyes with retinal fibrosis there is a lack of data considering focal retinal sensitivity to our knowledge [24,25,26]. Aggravating, there is a lack of uniformity considering the definition of retinal fibrosis and the ideal imaging method [27].

The objective of this study was to evaluate the effectiveness of different imaging techniques in detecting fibrosis through structure–function analysis.

## 2. Methods

### 2.1. Inclusion and Clinical Monitoring

The present cross-sectional, non-interventional study was conducted at the Department of Ophthalmology and Optometry of the Medical University of Vienna (Vienna, Austria).

A total of 30 eyes of 30 consecutive patients with nAMD and a history of at least one year of anti-VEGF therapy were analyzed. The study was approved by the local ethics committee (EK Nr: 1528/2012) and adhered to the tenets of the Declaration of Helsinki. Every patient gave written informed consent prior to study inclusion.

Inclusion criteria were a history of anti-VEGF treatment for a minimum of 1 year due to nAMD (treatment could be still ongoing or already paused or finished individually depending on the macula status of the patient), clear ocular media, and good fixation ability to guarantee appropriate image and testing quality. Exclusion criteria were: MNV due to diseases other than AMD, such as myopic MNV, other retinal disorders, uveitis, or clinically significant glaucoma.

In the case of nAMD in both eyes, the eye with the longer treatment duration was chosen.

Each patient underwent following examinations on the same day: Best corrected visual acuity (BCVA) on Early Treatment Diabetic Retinopathy Study charts (ETDRS), microperimetry (MP) (MP-3 Nidek, Aichi, Japan), slitlamp and dilated fundus examination, fluorescein angiography (FA) using the HR2 + OCT device (Spectralis, Heidelberg Engineering, Heidelberg, Germany), and SD/PS-OCT imaging, details of which are described below. All tests were performed by the same examiner (MS).

### 2.2. OCT Imaging

In this study, a custom-built spectral domain PS-OCT system, a functional extension of conventional SD-OCT, was used for imaging [28]. By exploiting the fact that some structures affect the polarization state of the light in a characteristic manner, additional tissue-specific contrast images can be retrieved.

The setup used in this study combines a PS-OCT system and a line scanning laser ophthalmoscope (LSLO) for retinal tracking in real time. The OCT operates at a center wavelength of 860 nm with a full width at half maximum bandwidth of 60 nm yielding an axial resolution of 4.2 μm (in tissue). The system operates at an A-scan rate of 70 kHz. The eye is illuminated by circularly polarized light and the backscattered light is detected at the interferometer exit by two identical custom-built spectrometers (i.e., one for each orthogonal polarization channel). The LSLO channel consists of a line illumination of the retina at a center wavelength of 785 nm and corresponding detection of the backscattered light by a line scan camera. Scanning of the LSLO illumination line over the retina is performed perpendicular to the illumination line (i.e., in y-direction) and the LSLO frame rate is 60 Hz. The LSLO images are used to determine lateral eye motion and corresponding offset values are sent to the x and y OCT galvanometer scanners for motion correction.

For each patient, a raster scanning over a region of 8 mm (x) × 6 mm (y) centered on the macula was performed. Each volume consists of 250 B-scans of 1024 A-scans each. Volume acquisition takes approximately 4.5 s. After the standard Fourier domain, OCT processing of the raw data and compensation of the corneal birefringence (Jones matrix-based algorithm), phase retardation, and optic axis orientation images (i.e., birefringence parameters) are computed in additionally to the usual intensity images. Averaged B-scans at specific positions (83 B-scans per position) were also acquired in order to increase the signal-to-noise ratio and consequently improve the polarization contrast. In addition to the birefringence parameters, the degree of polarization uniformity (DOPU) images are calculated (for both the 3D data and repeated B-scans). As the healthy RPE cell bodies constitute a depolarizing layer, they can accurately be identified and segmented from the DOPU images to perform further analysis [29].

### 2.3. Microperimetry

The automatic MP-3 (Nidek Japan) microperimeter, which contains a nonmydriatic fundus camera and has a dynamic range of 0–34 db, was used in this study.

Each examination was conducted after visual acuity testing and in mydriasis. Patients were briefed with exact instructions about the procedure and were supervised during their examination by an experienced examiner. Dark or light adaptation was not necessary prior to examination due to mesopic testing. In the beginning, the patient was seated and positioned correctly and comfortably, and the fellow eye was patched.

Refractive errors are automatically calculated and compensated by the MP-3 device.

The testing parameter settings were the following: a 45-stimuli grid within the central 10°, Goldmann III stimulus (0.43° diameter, which corresponds to 132 microns in the retina) in white with 200 ms duration time; 4–2 (fast) threshold strategy, and a red cross fixation target with a size of 1° and maximum intensity. The starting threshold was set to 17 db for one test location in each quadrant.

The stimuli grid was composed of 3 times 3 points within the central 2° area, 16 points within the 2° to 4° area, and 20 points within the 4° to 10° area. All spots fit within the fields of the EDTRS grid (see Figure 1).

The test was stopped automatically if severe eye movements were detected by the infrared eye tracker or manually if the patient required a pause.

At the end of the examination, CFP was performed automatically and overlaid with the IR image, which was taken at the beginning of the examination by the MP device. This co-registration was completed automatically but was manually in cases of significant discrepancies.

### 2.4. Image Analysis and Image Preparations

To prevent reader bias, the CFPs with the corresponding microperimetry grids were modified so that all sensitivity spots showed the same color independent of local retinal sensitivity (Figure 1). All MP spots on CFP were graded for the presence of fibrosis (yes or no) by reading center-approved retina experts (M.S./G.M.). In case of any disagreement, a third party (P.R.) was consulted, and a consensus was reached. Subretinal fibrosis was defined as a structure of fibrous-appearing tissue forming white or yellow plaques or mounds, which are well-defined in shape and might show signs of hyperpigmentation [4] (Figure 2A).

In the next step, corresponding PS-OCT and SD-OCT en face images were registered with the MP grid based on landmarks (i.e., retinal blood vessels) using a custom-built program (Matlab, Version: R2018a (MATLAB 9.4); The MathWorks, Inc., Natick, MA, USA) (see Figure 1). Subsequently, each stimulus location of the microperimetry grid was screened for fibrotic tissue that appeared as birefringent structures in the corresponding PS-OCT B-scans (retardation, axis orientation). The organization of collagenous fibers within subretinal fibrosis results in a column-like pattern in axis orientation B-scans, starting at the inner border of the fibrosis. The color within each column is rather homogeneous (see Figure 2H and Figure 3H) compared to non-fibrotic tissue which exhibits random noise (see Figure 2E and Figure 3E). The color differences between the columns originate from different angles of the axis orientation of the collagenous fibers with respect to the instrument [18]. The grading was performed by PS-OCT and retina experts (M.S./P.R.) and reviewed by experienced physicists (C.H. and M.P.).

In a third step, the SD-OCT images were evaluated for the presence of SHRM at the MP spot locations, by a reading center-proved retina expert (M.S.) (See Figure 2D,G and Figure 3D,G).

MNV classification was based on OCT and FA, which was performed at disease diagnosis and reviewed retrospectively based on consensus guidelines [30]. In type I MNV, the neovascular membrane is located between the retinal pigment epithelium (RPE) and Bruch’s membrane, in type 2 the lesion is above the RPE. Mixed type I and II MNV lesions show components of both types. RAP lesions (Type 3 MNV) show hyperreflectivity from the middle of the retina towards the RPE, together with intraretinal edema and bleeding.

### 2.5. Statistical Testing

For statistical testing IBM SPSS 25.0 was used.

Normally distributed descriptive statistics are reported as the mean ± standard deviation (SD). Not normally distributed descriptive statistics are reported as the median (1. quartile; 3. quartile). Independent samples *t*-test was performed to compare means, Mann–Whitney U test was performed to compare medians between groups.

Mixed linear models were used to find associations between retinal sensitivity in decibels measured by MP and corresponding local morphological findings, in the CFP images, the axis orientation/retardation B-scans, and the standard intensity-based OCT scans.

Using multiple mixed linear models, we calculated the association between retinal sensitivity in decibels, the presence of subretinal fibrosis on CFP, SHRM on OCT, and subretinal birefringence on PS-OCT separately for each parameter.

In each model the retinal sensitivity (in decibels) from microperimetry (MP) examinations was the dependent variable. As multiple sensitivity spots were tested per eye, the patient ID was implemented as a random factor into the model, to correct for multiple testing within a subject.

Additionally, we used a mixed model in which we calculated the association between retinal sensitivity and the number of image modalities (0,1,2,3) with fibrosis present (white or pigmented mounded lesion in CFP, SHRM in OCT and subretinal birefringence in PS-OCT) Pairwise comparison was Bonferroni corrected.

We corrected for the eccentricity of each sensitivity test point relative to the fovea and for the random factor patient eye, as multiple spots were evaluated per eye in each model.

The level of significance was set to *p* < 0.05.

## 3. Results

### 3.1. Patient Characteristics

A total of 30 eyes of 30 patients (15 female/15 male) with a mean age of 76.7 ± 6.3 years were included. The mean treatment duration prior to study inclusion was 3.7 ± 2.4 years, resulting in a mean of 18.0 ± 9.0 intravitreal anti-VEGF injections per patient (see Table 1).

A total of 14 (47%) patients also received anti-VEGF treatment for nAMD in the fellow eye.

Mean visual acuity was 60.2 ± 23.7 ETDRS letters. Thirteen (43%) eyes were pseudophakic, and 17 (57%) were phakic. Eighteen eyes had a type 1 (60%), 4 eyes a type 2 (13%), 1 eye a type 3 (3%), and 7 eyes a mixed type 1 and 2 (23%) MNV lesion.

The presence of a type 2 MNV component (type 2 MNV or mixed type MNV) was associated with a statistically significantly higher presence of subretinal fibrosis than a type 1 MNV (*p* < 0.05). Four patients presented with small (<1 papilla diameter) macular hemorrhages.

### 3.2. Retinal Sensitivity

A total of 1350 focal retinal locations (45 locations per study eye), were evaluated for retinal sensitivity on MP and analyzed for the presence of fibrosis on CFP, SHRM on intensity OCT scans, and subretinal birefringence on PS-OCT, respectively.

We found a significant correlation between impaired retinal sensitivity and the presence of fibrosis in CFP with an estimate of −13.08 db (*p* < 0.001), meaning that the retinal sensitivity is decreased by that amount. The estimated marginal means of this model were 12.60 db (95% confidence interval: 10.44–14.76) retinal sensitivity for a spot within an area of fibrosis based on CFP grading and 23.00 db (95% confidence interval: 20.96–25.04) retinal sensitivity for a spot within an area graded as non-fibrosis (Table 2).

Similarly, we found a significant correlation between impaired retinal sensitivity and the presence of SHRM on SD-OCT, with an estimate of −8.99 db (*p* < 0.001) for the presence of SHRM. The estimated marginal means of this model were 11.60 db (95% confidence interval: 8.84–14.36) for a spot within an area of SHRM based on OCT grading and 23.00 db (95% confidence interval: 19.62–24.63) for a spot without SHRM (Table 2).

We found a significant correlation between retinal sensitivity and the presence of subretinal birefringent tissue in PS-OCT with an estimate of −11.56 db (*p* < 0.001) for the presence of birefringent tissue. The estimated marginal means of this model were 11.02 db (95% confidence interval: 8.10–13.94) for a spot within an area of birefringence based on PS-OCT grading and 21.54 db (95% confidence interval: 19.05–24.03) for a spot within an area of non-birefringence (Table 2).

We found a significant correlation between retinal sensitivity and the number of imaging modalities showing signs of fibrosis, with an estimate of −16.9 db if fibrosis was detected in all 3 modalities (*p* < 0.001), −9.2 db if fibrosis was detected in 2 modalities (*p* < 0.001) and −9.2 db if fibrosis was detected in only 1 modality (*p* > 0.001). The estimated marginal means were 7.2 db if fibrosis was detected in 3, 10.1 db if detected in 2, 15.3 db if detected in 1 imaging modality, and 23.3 if no signs of fibrosis were found in either modality (See Table 3).

The pairwise comparison of this model showed significant differences between all pairs (*p* < 0.001) except for the comparison between 3 modalities and 2 modalities showing signs of fibrosis (*p* < 0.60). (See Table 4)

We implemented the amount of eccentricity of each tested spot relative to the fixation center (fovea) as a fixed factor into the mixed models and found that the corresponding estimate was positive in each model CFP: 0.29 db (*p* < 0.001), OCT: 0.46 (*p* < 0.001) and PS-OCT: 0.54 db (*p* < 0.001) implying that within this study population retinal sensitivity is higher with increasing distance to the fovea.

## 4. Discussion

CFP is currently the gold standard for detecting subretinal fibrosis, based on the appearance of mold-like whitish or hyperpigmented lesions. However, this modality may only depict the tip of the iceberg of retinal scarring. The use of OCT enables the evaluation of the retina on an almost histological level, in which fibrosis is represented volumetrically as hyperreflective material below the neurosensory retina (SHRM), accompanied by different amounts of RPE destruction and photoreceptor loss. However, SHRM is not solely associated with fibrosis, but can be the OCT correlate of multiple pathological findings associated with nAMD. These limitations rendered the use of PS-OCT imaging a valuable option in detecting retinal fibrosis, as this modality can identify collagenous fibers, the building blocks of retinal fibrosis. The downside of PS-OCT technology is its unavailability in most centers since these devices are not commercially accessible.

The aim of this cross-sectional study was to evaluate the suitability of three different imaging modalities (CFP, OCT, PS-OCT) in detecting fibrosis based on the correlation between morphological or structural changes of retinal fibrosis and retinal function, evaluated by MP sensitivity testing. Additionally, we evaluated whether multimodal imaging enhanced the identification of regions with localized reduction in retinal sensitivity, as a result of greater retinal damage due to fibrosis [31,32].

We found that retinal sensitivity was significantly reduced in areas graded as fibrosis in CFP (12.6 db), SHRM in OCT (11.6 db), or birefringent tissue in PS-OCT (11.0 db) compared to retinal sensitivity outside these locations (23.3 db), suggesting that all three modalities are suited for the detection of areas of decreased sensitivity. These fibrotic areas were more frequently found in eyes with a type II MNV, this is in line with the Harbor data [32].

CFP is a widely available tool and easy to use, however, differentiation between structures of similar clinical appearance is challenging and inherently subjective. AMD-related yellow-whitish material such as drusen or drusenoid pigment epithelial detachment (PED) may be misinterpreted as subretinal fibrosis on CFP but does not affect retinal sensitivity as much as fibrosis. We focused solely on nAMD and excluded other ocular diseases resulting in fibrosis, such as uveitis, due to the challenge of distinguishing AMD-related pathological findings from fibrosis [33]. To overcome this limitation, widely available conventional OCT systems are commonly used. In this imaging modality fibrosis appears as subretinal hyperreflective material (SHRM) and is easily distinguishable from healthy retina, PEDs, and drusen. Results of the Comparison of Age-related Macular Degeneration Treatments Trials (CATT) demonstrated that the prevalence of SHRM is high in eyes with untreated nAMD and decreases under anti-VEGF treatment over the years. Additionally, eyes with SHRM at baseline had lower VA regardless of its size or localization compared to eyes without SHRM. Also, the BCVA gain during follow-up was lower in eyes with persistent SHRM. A paper by Willoughby et al. also showed that consecutive scar development was significantly higher in eyes with persistent SHRM [4,9].

A recent study by Fang et al. came to similar results and showed that eyes with thinner SHRM were correlated with better vision [34]. Cheung et al. also showed a correlation between SHRM and reduced visual acuity [26].

These results beside other publications highlight the important role of SHRM in the development of fibrosis in nAMD and are in line with our findings, which show a decrease of retinal sensitivity in the presence of SHRM [35]. However, when it comes to alterations represented as SHRM in OCT such as MNV-membranes, hemorrhage, or fibrosis, a precise identification of the composition of SHRM or a differentiation between SHRM and underlying RPE is often not possible [9]. Therefore, a more specific imaging and analysis of SHRM is necessary to better understand the retinal scarring process.

To prove if a multimodal imaging approach could combine the strengths and reduce the weaknesses of the individual methods, we calculated a fourth mixed linear model, in which we evaluated the correlation between retinal sensitivity and the number of imaging modalities in which fibrosis was detected. The lowest mean retinal sensitivity (7.2 db) was observed in cases of signs of fibrosis in all three3 imaging modalities at the same spot, while the second lowest retinal sensitivity (10.1 db) was detected if signs of fibrosis were found in 2 imaging modalities. However, in the pairwise comparison, no significant difference was found between two or three imaging modalities (*p* = 0.06). If only 1 modality showed signs of fibrosis the retinal sensitivity was significantly better (15.3 db) compared to two or three modalities (*p* < 0.001) but significantly worse compared to a spot without any signs of fibrosis (23.3 db) (*p* < 0.001). The decrease in retinal sensitivity in fibrosis is expected and is caused by the loss of photoreceptors, as shown in the CATT study [31].

This result demonstrates that multimodal imaging is superior to a single-modality approach in detecting fibrosis. It also supports the findings of Toth et al. who highlighted the benefits of multimodal imaging, on a functional level [36]. The implementation of PS-OCT could improve fibrosis detection and give insight into other AMD-related findings, such as RPE alterations [37,38]. However, before PS-OCT systems can be commercialized corresponding image-processing algorithms need to be created.

A secondary finding of this study was the fact that all models showed a higher retinal sensitivity with an increase of eccentricity relative to the fixation center, the fovea. A possible explanation may be that AMD and nAMD are diseases primarily affecting the macular center. Hence, central retinal sensitivity would be most impaired compared to parafoveal areas.

Limitations of this study were the cross-sectional character, the relatively small sample size, and that a tested MP spot did not represent a single OCT A-scan but an area of approximately 130 microns within the retina which were covered by multiple A-scans within several B-scans. Additionally, despite eye-tracking technology, small eye movements during the MP examination may have had an influence on sensitivity levels especially in transition areas of healthy and unhealthy retina. A strength of this study was the multimodal imaging approach, including CFP and high-density axis orientation, retardation, and standard spectral domain OCT volumes, combined with microperimetry testing, which was to our knowledge never performed in eyes with fibrosis, resulting in a structure-function evaluation of retinal fibrosis.

## 5. Conclusions

In conclusion, our work suggests that each of the imaging modalities CFP, SD-OCT, and PS-OCT may be used individually to detect areas of reduced retinal sensitivity related to fibrosis. However, a multimodal imaging approach using at least two imaging modalities significantly improves the identification of areas of reduced retinal function. These areas are more likely to represent retinal fibrosis, which, alongside geographic atrophy, is the main cause of reduced visual acuity in AMD [39].

## Figures and Tables

**Figure 1 jcm-13-01074-f001:**
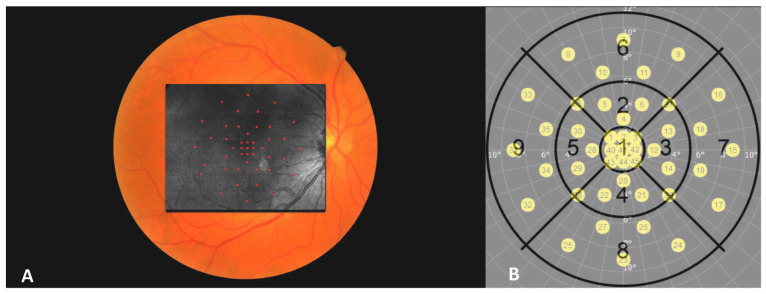
Image (**A**) shows the color fundus image and the registered pseudo-scanning laser ophthalmoscopy image with a overlaid microperimetry testing grid. Image (**B**) shows the microperimetry grid with all 45 test spots and the early treatment diabetic retinopathy study grid with corresponding degrees of excentricity relative to the center.

**Figure 2 jcm-13-01074-f002:**
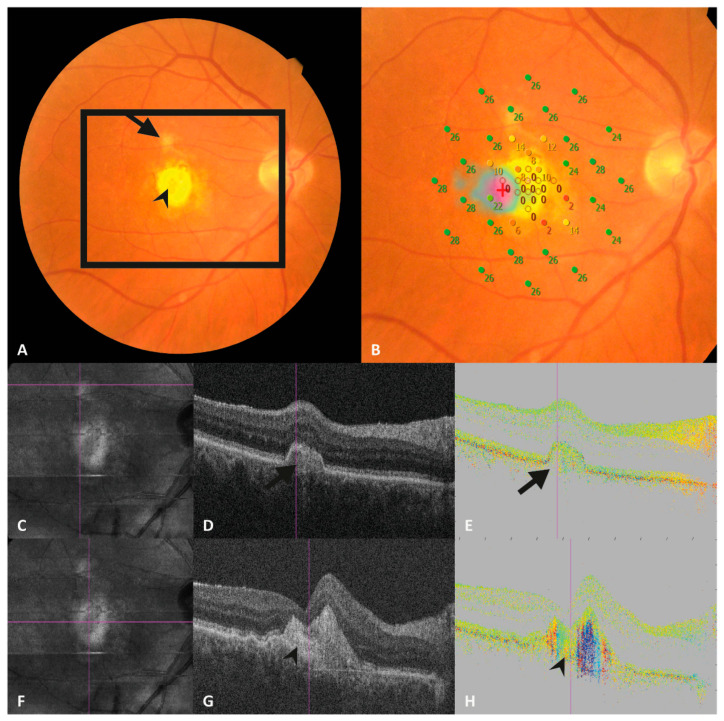
Right eye of a participant with a round white lesion on CFP (arrowhead, (**A**)) in the center of the macula and a smaller superior lesion (arrow, (**A**)). Retinal sensitivity given by microperimetry testing shows values of 0–10 db within the central lesion (**B**) whereas the spot of the microperimetry grid within the smaller lesion shows a sensitivity of 26 db (**B**). (**C**,**F**) show the pseudo-scanning laser ophthalmoscope images calculated from SD-OCT volume data of the PS-OCT, the violet lines indicate the location of the corresponding intensity optical coherence tomography (OCT) B-scans (**D**,**G**) and the polarization-sensitive OCT axis orientation scans (**E**,**H**). (**D**) shows a pigment epithelial detachment co-localized with the smaller white lesion on CFP (arrow, (**A**)). The corresponding PS-OCT scan (**E**) shows no birefringence indicating a non-fibrotic lesion. Image (**G**) represents an OCT scan of the central lesion (arrowhead, (**A**)) showing subretinal hyperreflective material. The corresponding PS–OCT scan shows birefringent material indicating the presence of subretinal fibrosis.

**Figure 3 jcm-13-01074-f003:**
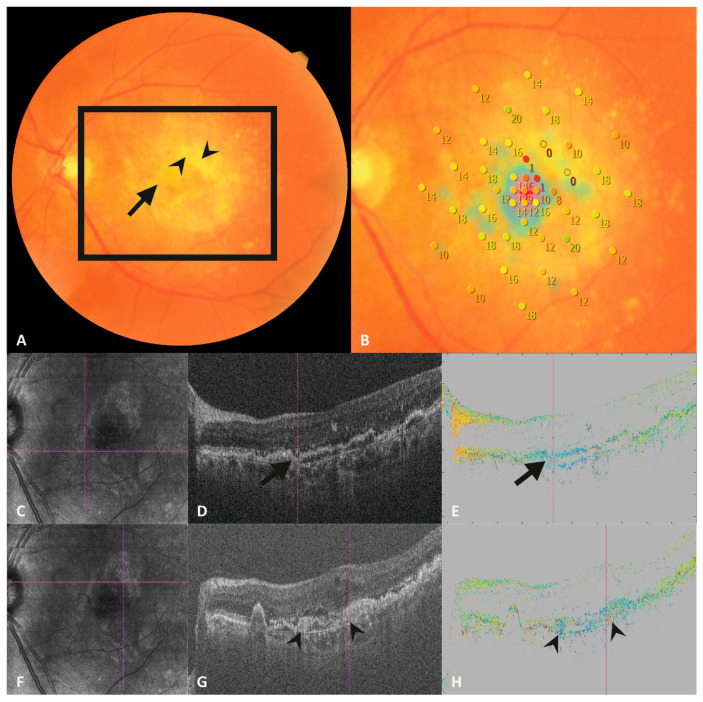
Left eye of a participant with a large diffuse white lesion, with areas of homogeneous white subretinal material (arrowhead, (**A**)) and areas of irregularly distributed white subretinal material (arrow, (**A**)). Retinal sensitivity on microperimetry shows values of 0–1 db within homogeneous white areas (**B**), and values of 8–16 db within the irregular areas of the lesion (**B**). (**C**,**F**) show the pseudo-scanning laser ophthalmoscope images calculated from SD-OCT volume data of the PS-OCT, the violet lines indicate the location of the corresponding intensity optical coherence tomography (OCT) B-scans (**D**,**G**) and the polarization-sensitive OCT axis orientation scans (**E**,**H**). (**D**) shows the OCT B-scan in the area of the lesion with better retinal sensitivity (arrow, (**A**)) appearing like a drusen (arrow, (**D**)). The corresponding PS-OCT scan shows no birefringence (**E**). (**G**) shows an OCT B-scan of an area of homogeneous white subretinal material (arrowhead, (**A**)) showing Drusen and RPE irregularities mixed with subretinal hyperreflective material (arrowheads, (**G**)). The corresponding PS-OCT scan shows two small columns of birefringent material (arrowheads, (**H**)), representing collagen fibers of retinal fibrosis.

**Table 1 jcm-13-01074-t001:** Characteristics of study participants. The number of anti-VEGF injections contains Aflibercept and Ranibizumab. Absolute and (relative) amount of each macular neovascularization (MNV) type within the study population.

General Data	
Number of patients	30 (15 males, 15 females)
Mean BCVA	60.2 ± 23.7 letters
Age	76.7 ± 6.3 years
Treatment Duration	3.7 ± 2.4 years
Numbers of anti-VEGF injections	14.3 ± 6.9 injections
MNV in the fellow eye	14 patients (46.7%)
MNV type I	18 (60%) eyes
MNV type II	4 (13.3%) eyes
MNV type I + II	7 (20.3%) eyes
MNV type III	1 (3.3%) eye

**Table 2 jcm-13-01074-t002:** Shows the results of 3 different mixed linear models.

Color Fundus Photography			Standard OCT			Polarization Sensitive OCT		
Parameter	Estimate	*p*-Value	Parameter	Estimate	*p*-Value	Parameter	Estimate	*p*-Value
Constant term	+21.5	0.349	Constant term	+19.41	<0.001	Constant term	+18.75	<0.001
Degree of center	+0.29	<0.001	Degree of center	+0.46	<0.001	Degree of center	+0.54	<0.001
Presence of fibrosis	−13.08	<0.001	Presence of SHRM	−8.99	<0.001	Presence of Bi-refringence	−11.56	<0.001
Interaction Presence of fibrosis * degree of center	+0.52	0.001	Interaction Presence of SHRM and degree of center	−0.24	0.258	Interaction Bi-refringence and degree of center	+0.20	0.428

* The “degree of center”-parameter, is a value for the distance between a tested MP spot and the fixation target (the fovea) in degrees. The level of significance was set to 0.0167 using Bonferroni correction. SHRM = subretinal hyperreflective material, PED = pigment epithelial detachment, Bi-refringence = a finding in polarization-sensitive OCT, which is at the subretinal level pathognomonic for retinal fibrosis.

**Table 3 jcm-13-01074-t003:** Shows the Estimated marginal means of a mixed linear model calculating the correlation between retinal sensitivity in decibels and the number of modalities in which fibrosis was detected. Estimated marginal means modality mixed models.

	Mean (Decibel)	95% Confidence Intervall (Decibel)
Fibrosis in 3 modalities	7.2	4.6–9.8
Fibrosis in 2 modalities	10.1	7.7–12.4
Fibrosis in 1 modality	15.3	13.2–17.4
Fibrosis in 0 modality	23.3	21.4–24.2

**Table 4 jcm-13-01074-t004:** Pairwise comparison of a mixed linear model calculating the correlation between retinal sensitivity in decibels and the number of modalities in which fibrosis was detected. Pairwise comparison modality mixed models.

		Difference of Means	Significance
Fibrosis in 3 modalities	Fibrosis in 2 modalities	−2.9	0.060
	Fibrosis in 1 modality	−8.1	<0.001
	Fibrosis in 0 modality	−16.1	<0.001
Fibrosis in 2 modalities	Fibrosis in 3 modalities	2.9	0.060
	Fibrosis in 1 modality	−5.2	<0.001
	Fibrosis in 0 modality	−13.2	<0.001
Fibrosis in 1 modality	Fibrosis in 3 modalities	8.1	<0.001
	Fibrosis in 2 modalities	5.2	<0.001
	Fibrosis in 0 modality	−8.0	<0.001
Fibrosis in 0 modality	Fibrosis in 3 modalities	16.1	<0.001
	Fibrosis in 2 modalities	13.2	<0.001
	Fibrosis in 1 modality	8.0	<0.001

## Data Availability

The data presented in this study are available on request from the corresponding author (accurately indicate status).

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
