# Peer review of "Structure-Function Correlation of Retinal Fibrosis in Eyes with Neovascular Age-Related Macular Degeneration"

_jcm, 2024, doi:10.3390/jcm13041074_

Round 1

Reviewer 1 Report

Comments and Suggestions for Authors

In this manuscript, the authors summarized and discussed the structure-function correlation of retinal fibrosis in eyes with neovascular age-related macular degeneration. On this basis, they further investigated the structure function correlation of subretinal fibrosis using retinal microperimetry, color fundus photography, optical coherence tomography and polarization-sensitive optical coherence tomography imaging. This work provides new insight and opinion into the development of different imaging methods to detect areas of reduced retinal sensitivity related to fibrosis. The manuscript is well-organized and clearly stated. I have following minor concerns to be addressed.

1.      There are two repetitions of conclusion in lines 28 to 35 of the original manuscript.

2.      Line 21 “SD-”, I guess the author means “SD-OCT”.

3.      Line 333 “(11.0)”, I guess the author means “(11.0db)”.

4.      Line 363,365,368,370, I noticed that sometimes it is (xxdb), and sometimes it is (xxdB). Is it necessary to unify it?

5.      I wonder if 30 patients can provide sufficient evidence for this type of research.

6.      In the conclusion part, more relevant documents can be cited to support the viewpoint.

Comments on the Quality of English Language

Apart from some details in case, I think the quality of English in this paper is all right.

Reviewer 2 Report

Comments and Suggestions for Authors

Thank you for allowing me to review the manuscript entitled “Structure-Function Correlation of Retinal Fibrosis in Eyes with Neovascular Age-Related Macular Degeneration”. This paper focuses on the the structure function correlation of subretinal fibrosis in eyes with neovascular AMD.  I could not understand the purpose of this study. Below I discuss some issues, which hopefully can help you improve the study. 

Abstract: Authors mentioned conclusion twice.

Abstract: Purpose and conclusion do not match.

Please explain in more detail the inclusion criteria for this study (i.e. under treatment?, duration from last injection). 

Did all eyes included in this study involve the fovea?

l105-106, Were there any cases with inability of fixation due to poor visual acuity in consecutive patients?

Were all tests performed on the same day?

The significance of identifying the location of fibrosis is not clear. There are other causes of reduced retinal sensitivity besides fibrosis.

There are two graders in this study. How did authors decide when there was a difference of opinion?

SHRM detected by OCT does not necessarily represent fibrosis. Did any of the cases in this study have hemorrhage?

l394-395 Areas of low sensitivity/ retinal function do not necessarily represent fibrosis.

Is this a study to better understand if the decreased sensitivity is due to fibrosis? Or, since we know that area of fibrosis are associated with reduced retinal sensitivity, is this a study to detect the location where fibrosis is occurring or occurred?

Reviewer 3 Report

Comments and Suggestions for Authors

Basing on the limitations of current ophthalmic examination methods such as retinal microperimetry (MP), fundus color photograph (CFP), and optical coherence tomography (OCT) , the article aims to use multiple methods of examinations incluing polarization sensitive optical coherence tomography (PS-OCT) in the evaluation of subretinal fibrosis. The patients with neovascular age-related macular degeneration (nAMD) were enrolled and performed MP, CFP, OCT and PS-OCT to comprehensively evaluate subretinal fibrosis, in order to find a more sensitive and accurate evaluation model for subretinal fibrosis. It is interesting. However, some modifications should be made。

The following related questions are as follow:

1 Only 30 patients were selected in the experiment, and the sample size was too small, making the statistical conclusions prone to bias. The authors need to detail the exclusion criteria and inclusion criteria in the Methods section.

2. A total of 30 eyes and 30 patients were selected in the experiment. Is there only one eye tested for each patient? Or test the more serious eye in both eyes?

3. Are all the experimental tests, such as OCT, PS-OCT, MP, and CFP, done by the same doctor? If not, how can we exclude the artificial influencing factors in the test results?

4 The previous part of this article has been published in IOVS. What are the innovations in this study compared with the previous results? The purpose of this study was to evaluate the correlation between the structure of subretinal fibrosis and retinal function by combining a variety of examination methods. Why only nAMD patients were included in the study, and what were the exclusion criteria for other ocular diseases with subretinal fibrosis?The authors might elucidate it in the discussion section.

5 This study proves that the sensitivity and accuracy of a multimodal examination method are better than that of a single examination method, but currently PS-OCT cannot be popularized and widely used. Is it of clinical value to use this examination to evaluate subretinal fibrosis in nAMD patients? It is suggested that the author look into the application prospect in the discussion.

Round 2

Reviewer 2 Report

Comments and Suggestions for Authors

I think that all of the issues have been addressed.

Reviewer 3 Report

Comments and Suggestions for Authors

The reversion is well organized.